# Prediction Models Based on Regression and Artificial Neural Network for Moduli of Layers Constituted by Open-Graded Aggregates

**DOI:** 10.3390/ma14051199

**Published:** 2021-03-04

**Authors:** Yunje Lee, Yongjin Choi, Donghyun Ahn, Jaehun Ahn

**Affiliations:** Department of Civil and Environmental Engineering, Pusan National University, Busan 46241, Korea; lee_yunje@pusan.ac.kr (Y.L.); yj.choi@pusan.ac.kr (Y.C.); Civi0641@naver.com (D.A.)

**Keywords:** permeable pavement, modulus of elasticity, open-graded aggregate, Plate Load Test, Light-Weight Deflectometer, Linear Regression, Artificial Neural Network

## Abstract

The impermeable cover in urban area has been growing due to rapid urbanization, which prevents stormwater from being naturally infiltrated into the ground. There is a higher chance of flooding in urban area covered with conventional concretes and asphalts. The permeable pavement is one of Low-Impact Development (LID) technologies that can reduce surface runoff and water pollution by allowing stormwater into pavement systems. Unlike traditional pavements, permeable pavement bases employ open-graded aggregates (OGAs) with highly uniform particle sizes. There is very little information on the engineering properties of compacted OGAs. In this study, the moduli of open-graded aggregates under various compaction energies are investigated based on the Plate Load Test (PLT) and Light-Weight Deflectometer (LWD). Artificial Neural Network (ANN) and Linear Regression (LR) models are employed for estimation of the moduli of the aggregates based on the material type and level of compaction. Overall, the moduli from PLT and LWD steeply increase until the number of roller passes reaches 4, and they gradually increase until the number of roller passes becomes 8. A set of simple linear equations are proposed to evaluate the moduli of open-graded aggregates from PLT and LWD based on the material type and the number of roller passes.

## 1. Introduction

The rainfall pattern changes globally due to climate change, and therefore the rainfall can be much more intense than in the past. In addition, the impermeable cover in urban areas has been growing due to rapid urbanization [1,2]. Owing to climate change and urbanization, the cities suffer from heavier flood damages nowadays. Low-Impact Development (LID) is a design philosophy that can remedy these issues, often using surface-infiltrating stormwater facilities, and is being widely implemented worldwide [1,2,3,4,5]. The permeable pavements are one of the most popular LID technologies which can especially be well-applied in urban areas [6,7,8,9,10,11,12], which are implemented to not only low-traffic local roads but also high-traffic highways. Figure 1 shows typical sections of permeable pavements. Permeable pavements bring other benefits such as reducing the heat island phenomenon, leveling down traffic noise, and providing better skid resistance [9].

A permeable pavement has a surface layer that is in direct contact with the vehicle load. In addition, it has a base layer supporting the surface layer that disperses the vehicle load transmitted through the surface layer [6,8,10]. Compaction of this base layer during road construction plays a significant role in the stability and longevity of the pavement [13,14]. Therefore, proper compaction quality control of the base is essential for the construction of roads to ensure that they meet the required engineering specifications. However, a standard for the compaction quality control of open-graded aggregates (OGAs), which are used to construct base or sub-base layers of permeable pavements, has not been established, despite the increase in the applicability of permeable pavements [6,8,9,10,11,12]. In addition, the experimental study conducted in the lab environment does not necessarily reflect what would happen in the field. The magnitude of compaction energy and the way that the energy is delivered cannot be the same in the field and lab. In this study, the moduli of OGAs of various sizes under various compaction energies are investigated based on PLT (Plate Load Test) and LWD (Light-Weight Deflectometer). There have been very few experimental studies on the stiffness of open-grade aggregates in field conditions. Artificial Neural Network (ANN) and Linear Regression (LR) models are employed for estimation of the moduli of the aggregates based on the material type and level of compaction.

## 2. Field Modulus Evaluating Devices

In recent years, due to the implementation of mechanistic-empirical pavement design procedures (MEPDG) [15], modulus-based compaction specifications have been established [16], and many kinds of field modulus evaluating devices have been developed and used to evaluate the field compaction quality of pavement base layers [17]. These include test methods and devices such as PLT [18], LWD [19], soil stiffness gauge (SSG) [20], and dynamic cone penetrometer (DCP) [21].

Various studies regarding these devices or test methods have been conducted focusing on conventional road embankment construction materials. These studies have explored the feasibility of LWD, SSG, and DCP in compaction quality control during the construction based on laboratory or field experiments [13,14,17,22,23,24,25,26,27,28], investigated the variation of modulus in response to density or moisture content changes [24,26,27,28,29], compared modulus measurements from each device and suggested relationships between them [14,23,26], and investigated statistical limits of the measurements in construction quality assurance [30]. In this study, we coped with the modulus measurement data obtained from PLT and LWD tests.

The PLT is a test that evaluates the mechanical properties of the ground based on the relationship between the load and the settlement after applying a load through the rigid loading plate to the ground where the structure is installed. It is used to evaluate the stiffness of roads and the ground [31]. The test is divided into the non-repetitive loading method and the repetitive loading method depending on the loading. In Korea, the Korean Standards Association’s KS F 2310 [32] applies to the non-repetitive loading method. The German Standardization Association’s DIN 18134 [18] standard applies to the repetitive method, which is used in this study. There are loading and unloading stages in the test. From each stage, a stress-settlement curve is obtained by the second-order polynomial regression, and modulus is determined from the curve, which is represented as E_v1_ (primary strain modulus) and E_v2_ (secondary strain modulus), respectively. The configuration of the test equipment is shown in Figure 2.

The LWD measures modulus by dropping a falling weight at a certain height to its loading plate and measuring the maximum settlement of it. E_LWD_ is the elastic modulus evaluated using the LWD. The elastic modulus is one of the indicators of the stiffness of the ground. It is a method to evaluate the compaction of the ground. Compared to the PLT, this test method has the advantage of being a simple test method with a short test time and portability [17]. The configuration of the test equipment is shown in Figure 3. A total of six drops should be performed to complete one test. The first three drops are steps to properly set the loading plate on the ground, and the latter three are steps to measure the settlement amount of the loading plate. The settlement of the loading plate is measured for each drop over time through an accelerometer mounted on the loading plate. The average value of the last three settlement measurements is used to calculate E_LWD_.

Several studies have been conducted using the PLT and LWD test to evaluate the modulus or compaction of conventional road base construction materials, which are dense-graded aggregates. Kim and Park [23] and Kim et al. [35] conducted PLTs on pavement base layers based on both German standard, DIN 18134 [18], and KS F 2310 [32], and the results from both standards were compared. Wiman [36] evaluated the permanent deformation and modulus of base materials through the LWD and PLT based on DIN 18134 [18]. Berney, Mejais-Santiago, and Kyzar [29] constructed testbeds composed of various dense-graded base materials to evaluate the modulus of them using the LWD with other devices such as SSG and DCP and accumulated statistical data of the measurements.

The modulus-based compaction quality control technique can be an effective means for evaluating the compaction quality of OGAs because it is difficult to measure the maximum dry density and on-site dry density of them. However, most research about modulus-based compaction quality control has been focused on the materials used in conventional road base materials, except for a few studies conducted by Choi [37] and Choi et al. [38].

## 3. Field Experiment

### 3.1. Test Materials

The base layer in a road pavement is installed directly below the surface layer, and it distributes the traffic load received from the surface layer [39]. Since it is located under a thin surface layer, the load pressure distribution transmitted to the base layer is high; therefore, the selected base layer material needs to support the load and have sufficient resistance to deformation. In addition to these properties, the base in the permeable pavement should have a hydrological function to allow water to penetrate and retain rainwater in the ground. To this end, the base of the permeable pavement is composed of OGAs having an even particle size and almost no particulates, unlike the base layer of a road pavement composed of dense-graded aggregates.

In the field test, three different kinds of OGAs were used: D40 (maximum particle size 40 mm), D25 (maximum particle size 25 mm), and D13 (maximum particle size 13 mm). These particle sizes approximately lie in a commonly used particle size range of the base materials of permeable pavements. The aggregates were prepared in an air-dried state with about 0.5% of water contents. D40 and D25 were mixed 1:1 to make D40 + D25, and D25 and D13 were mixed 1:1 to make D25 + D13. A backhoe was used for the mixing of the aggregate, and a backhoe bucket was used to measure the volume of those aggregates to be mixed. In the field test, a total of five types of aggregates were prepared as test materials and were used in the construction of the test site. The aggregate volume composition, basic properties, and particle size distribution of each material are shown in Table 1 and Figure 4. In addition, the particle size distributions of OGAs used in typical permeable pavement base layers suggested by Smith [6] and Eisenberg, Lindow, and Smith [8], which is ASTM No. 57 aggregates, and Seoul Metropolitan City [10] are plotted together for comparison. Most of the materials coped within this study showed similar particle size distributions with the suggested range, with some exceptions in D13.

### 3.2. Test Program

The test was planned as follows using the five aggregates described above. In order to investigate the change in the stiffness of the compacted aggregate layer depending on the number of roller passes performed using a compaction roller, a field testbed was constructed following the dimensions illustrated in Figure 5. Each aggregate was placed to a height of 30 cm on the road (first lift), and then compacted 2, 4, 8, and 12 times with a 10 ton vibration compaction roller. Two PLTs and four LWD tests were then conducted at each number of compaction (roller passes) for all materials. A 30 cm diameter plate was used for the test. After all these tests were completed, an additional layer of each aggregate was laid to a depth of 30 cm on top of the previously laid aggregate layer (second lift), and the same measurement process was repeated. As a result, a total of 80 PLT and 160 LWD measurements were carried out throughout the whole test. Table 2 summarizes the whole test program.

Figure 6 shows the locations corresponding to the test measurements. In the figure, the gray circles indicate the PLT measurements which have been conducted twice for each batch. LWD tests were conducted twice for each PLT test, a total of four times each batch. The numbers 2, 4, 8, and 12 indicate the number of roller passes, and A and B indicate the first and second PLT test positions. This layout of test location was applied to the tests for all materials.

## 4. Modulus of Open-Graded Aggregate

### 4.1. Field Test Results

Table 3 presents 80 cases of datasets including types of materials, the number of roller passes, and the moduli evaluated using PLT and LWD. E_LWD_ is the elastic modulus measured with the LWD test and E_v1_ and E_v2_ are the primary and secondary strain moduli, measured with the PLT test respectively. For each PLT, LWD tests were conducted twice, and therefore E_LWD_ in the table is the average of two test results. There are five types of materials: D40, D40 + D25, D25, D25 + D13, and D13. Each of the materials is replaced with a numerical value of 1, 2, 3, 4, and 5, subsequently, to use as an input to the model. Areas A and B in Table 3 represent the test locations for PLT shown in Figure 5. In order to capture trends of the data in Table 3, it is also plotted in Figure 7, Figure 8 and Figure 9.

The zone of influence of PLT is known as 1.5 to 2 times the diameter of the plate in the literatures based on experimental and numerical studies [41,42]. The zone of influence of LWD, on the other hand, may be identical to or a bit shallower than that of PLT. The zone of influence of LWD is reported to be 1.5 to 2 times [28] or 1 to 2 times the diameter [43,44]. Field tests have been conducted on the well-compacted and stiffer ground. Following the zone of influence, the test results for the first lift could possibly be influenced by the stiff bottom layer. However, when the moduli for the first and second lifts are compared, the results of the first lifts are not consistently higher than those of the second lifts. The open-graded aggregates tested in this study have much higher porosity and set-up with fewer particle-to-particle contacts than dense materials. The aggregates went through highly plastic behavior. Authors suspect the zone of influence was contained within a shallow depth with high plastic deformations. As there was no consistent difference found in the results of the first and second lifts, the number of the lift was not used as a variable in the regression and neural network analyses.

### 4.2. Modulus Results Depending on the Number of Roller Passes

The moduli E_v1_, E_v2_, and E_LWD_ are plotted with respect to the number of roller passes and material type in Figure 7, Figure 8 and Figure 9 with the data presented in Table 3. Both the scatter plot and surface plot, which consists of the average values of the data points of the scatter plot, are shown in those figures to better visualize data. The E_v1_ and E_v2_ values steeply increase until the number of roller passes reaches 4. After 4 passes, E_v1_ continues to increase with increasing number of roller passes with less steep rate, which was not the case for E_v2_ and E_LWD_. During the first loading, the specimens are compressed and densified (or compacted), and therefore the second compression curve (or E_v2_) is less sensitive to the number of roller passes. The first loading of the Plate Load Test also involves significant shear displacement, which does not happen during the second static loading or dynamic loading: E_v1_ is more sensitive to the number of roller passes than E_v2_ and E_LWD_ overall. All three moduli, E_v1_, E_v2_, and E_LWD_, values nearly settle after the number of roller passes becomes 8. The number of roller passes 4, which is typically used in practice, may be an efficient number, but one may consider applying the number of roller passes 8 to achieve better stiffness for open-graded aggregates. The difference in measurements with LWD and PLT are due to different principles assumed for evaluation of the moduli. Therefore, the differences in measurement are owing to influence depth, loading rate, and the number of loading postulated for each evaluation method.

## 5. Artificial Neural Network and Linear Regression

It was attempted to evaluate the moduli of open-graded aggregates based on either ANN or Linear Regression [45,46,47,48], in order to see whether the moduli can be predicted well from aggregate and compaction information. MATLAB [49] and MS Excel [50] were used to implement ANN and LR, respectively.

### 5.1. Artificial Neural Network Model

The human brain solves many problems that are difficult to solve using current information processing technologies. Studies have been performed to analyze and model the functional factors of the human brain that entail excellent thinking ability, memory, problem analysis, and solving ability [46,51]. Machine learning is a technology through which a computer learns from accumulated data, and it is one of the artificial intelligence technologies developed in the 1980s. The algorithms used for machine learning include decision trees, clustering algorithms, and ANNs. Of these, an ANN operates in a manner similar to a neural network structure in which multiple neural cells connected to each other share signals and there is no direct connection between the input layer and the output layer. An ANN is divided into an input layer, which receives data as inputs, hidden layers that represent a complex relationship between the input and output, and the output layer, which produces the final result (Figure 10) [47,48,51].

The data used to train the ANN were the material type, the number of roller passes, and the moduli measured via PLT and LWD. Two different ANN models, ANN1 and ANN2, were set and applied. The “ANN1” model entailed the material type and the number of roller passes as features (inputs), and the moduli evaluated via PLT and LWD as labels (outputs), as presented in Figure 11. Therefore, in ANN1, it is attempted to predict the moduli based only on the material size and compaction level. The “ANN2” model, on the other hand, incorporates the modulus from LWD as a feature, not a label—the labels are the moduli from PLT (Figure 12). PLT generally costs much more time and effort to conduct than LWD. If one can successfully evaluate the results of PLT based on those of LWD, it would simplify the process of quality assessment while still keeping the benefits of PLT. An ANN structure with one hidden layer is selected, and 20 nodes were used. To determine the number of nodes, several different values of nodes were tried, and the use of 20 nodes resulted in efficient performance.

### 5.2. Linear Regression Model

Simple LR models were implemented to account for the relationships between the features and labels which have been postulated in ANN1 and ANN2 models (Figure 11 and Figure 12). ANN is the black box model, but with LR, explicit equations can be developed for inputs and outputs enabling an easier application to practice. Two LR models were postulated, LR1 and LR2, which employ the same features and labels for ANN1 and ANN2, respectively. LR1 entails the material type (1, 2, 3, 4, 5) and the number of roller passes (2, 4, 8, 12) as inputs, and the moduli from PLT and LWD as outputs (see Figure 13), while LR2 takes the modulus from LWD as an input, not an output (see Figure 14). As such, the completed LR1 model equates as follows:E_LWD_ = −3.55 N_mat_ + 0.7 N_rp_ + 37.01(1)
E_v1_ = −0.9 N_mat_ + 0.93 N_rp_ + 11.53(2)
E_v2_ = −2.64 N_mat_ + 1.7 N_rp_ + 115.52(3)
where E_LWD_ is the modulus from LWD, E_v1_ and E_v2_ are the moduli from PLT for loading and reloading, N_mat_ is the material type, and N_rp_ is the number of roller passes. On the other hand, the LR2 model formulates as follows:E_v1_ = −0.33 N_mat_ + 0.75 N_rp_ − 0.18E_LWD_ + 5.46(4)
E_v2_ = −3.5 N_mat_ + 1.77 N_rp_ − 0.29E_LWD_ + 127.29(5)

Root mean square error (RMSE) values of the equations presented above are presented in Table 4 and Table 5.

### 5.3. Evaluation Results and Comparison

The experimental data had a total of 80 sets, as presented in Table 3, and 70%, 15%, and 15% of data were randomly selected and used for training, validation, and testing (prediction) respectively, with an option implemented in MATLAB. The results of the evaluation of the models ANN1 and LR1 are presented in Figure 13 together with the baseline values (experimental results). The horizontal axis in the figure represents the dataset not used for training of the ANN model or fitting of the LR model (saved for evaluation of model performance). It is noted that the prediction based on the ANN model does not necessarily provide a better match with the baseline. For some datasets, LR prediction is better; for some others, ANN gives better prediction. In Figure 14, the results of ANN2 and LR2 models are shown. For amodels, ANN2 and LR2, for some datasets, the ANN2 model predictions are closer to the baselines, but for others, LR2 predicts closer values to the baselines.

RMSE (root mean square error) of the ANN and LR models were estimated to investigate overall performance of prediction of proposed models. RMSE is a statistical measure of the difference between the predicted and baseline, and can be equated as follows:(6)$$\mathrm{RMSE}\;=\;\sqrt{\frac{1}{n}{\left(M_{i}-P_{i}\right)}^{2}}$$
where *M_i_* is the predicted, and *P_i_* is the baseline.

Table 4 presents the RMSE values for training and testing of ANN1 and LR1 (see Figure 13). In the case of the model LR1, for all three moduli, E_LWD_, E_v1_, and E, the RMSE values for training (fitting) and testing (prediction) are quite close; during fitting and evaluation, the LR1 model produces similar level of errors. In the model ANN1, on the other hand, for E_LWD_ and E_v2_, the RMSE for testing is larger than that for training. When the testing RMSEs of two different models, ANN1 and LR1, are compared, RMSE of ANN1 is higher for E_LWD_, but RMSE of LR1 is higher for E_v1_. One model is not necessarily superior to the other for prediction. The results of the model evaluations of ANN2 and LR2 (see Figure 14) are presented in Table 5. The ANN2 model works better for training (or fitting), but the LR2 model is slightly better at testing (prediction).

PLT costs more much more time and efforts to conduct than LWD and replacing PLT by LWD would make the quality assessment procedure simpler. The intention of models ANN2 and LR2 is to predict the results of PLT based on those of LWD. While the models ANN1 and LR1 employ only the material type and the number of roller passes as inputs, the models ANN2 and LR2 additionally have E_LWD_ as an input. However, by comparing Table 4 and Table 5, it can be seen that the models ANN1 and LR1, which have fewer inputs, therefore, are simpler, and have better perdition overall, especially for E_v2_. It appears that there is no significant correlation between the moduli from PLT and LWD, as these two methods evaluate the moduli on entirely different bases.

Continuous Compaction Control (CCC) or Intelligent Compaction (IC) refers to an innovative compaction technique which can perform field compaction simultaneously evaluating the stiffness of soil, and therefore compaction quality [52]. There was no vibration sensor implemented to the compaction roller in this study, thus CCC measurements are not available for comparison with LWD and PLT measurements. However, when CCC or IC is applied to the open-graded aggregates, the stiffness results of LWD and PLT and regression equations presented in this study may be taken as references, making CCC operations more reliable.

## 6. Summary and Conclusions

In this study, the moduli of open-graded aggregates under various compaction energies were investigated based on PLT (Plate Load Test) and LWD (Light-Weight Deflectometer). Artificial Neural Network (ANN) and Linear Regression (LR) models were employed for estimation of the moduli of the aggregates based on the material type and level of compaction. The conclusions obtained are as follows:The modulus from the first loading curve of PLT was more sensitive to the number of roller passes than the moduli from the reloading curve of PLT and from LWD. It is due to the significant compressional and shear deformation that happens during the first loading of PLT, which does not appear during the reloading of PLT and dynamic loading of LWD.Overall, the moduli from PLT and LWD steeply increase until the number of roller passes reaches 4, and they gradually increase until the number of roller passes becomes 8. The number of roller passes 4, which is typically used in practice, may be an efficient number, but one may consider applying the number of roller passes 8 to achieve better stiffness for open-graded aggregates.The models that do have one less input (the modulus from LWD) actually performed a little better than the models with more inputs. When there is no good correlation between input and output (e.g., the moduli from PLT and LWD), adding more input variables does not necessarily help the prediction of the model.A set of simple linear equations were proposed to evaluate the moduli of open-graded aggregates from PLT and LWD based on the material type and the number of roller passes. The predictions based on ANN models did not necessarily provide a better match with the baseline compared to LR models.The stiffness results of LWD and PLT and regression equations presented in this study may be taken as references when CCC operations are made in open-graded aggregates. The characteristics of open-graded aggregates of different sizes, shapes, and minerals should be further studied for reliable application of open-graded aggregates to the base of permeable pavements.

## Figures and Tables

**Figure 1 materials-14-01199-f001:**
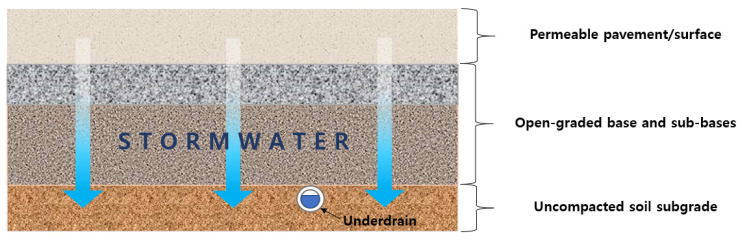
Permeable pavements.

**Figure 2 materials-14-01199-f002:**
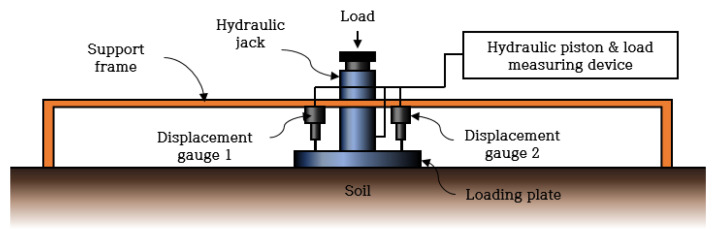
Plate Loading Test (PLT) equipment composition [33].

**Figure 3 materials-14-01199-f003:**
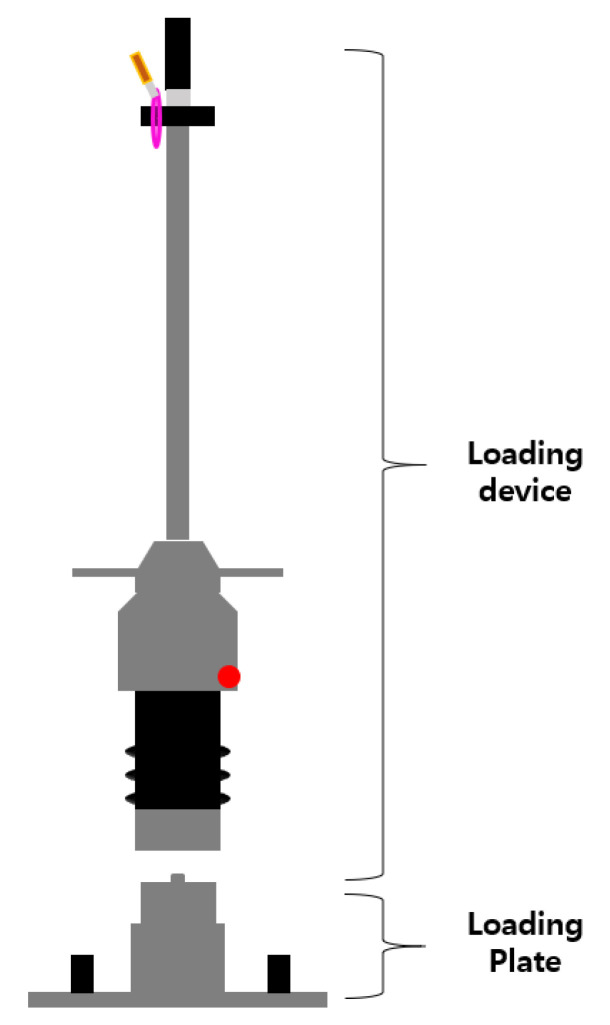
Light-Weight Deflectometer (LWD) equipment composition [34].

**Figure 4 materials-14-01199-f004:**
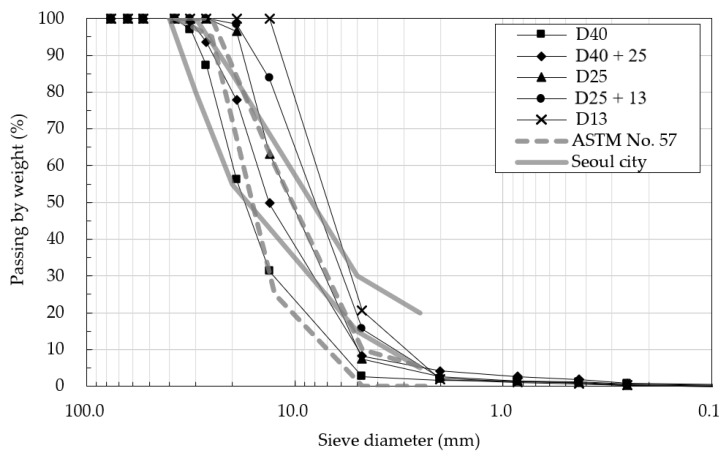
Particle size distribution of test materials and specifications [38].

**Figure 5 materials-14-01199-f005:**
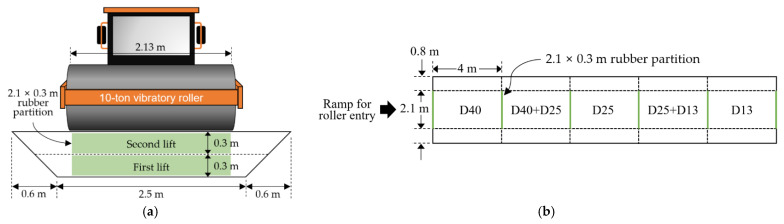
Testbed layout [38]. (**a**) Cross-sectional view; (**b**) plan view.

**Figure 6 materials-14-01199-f006:**
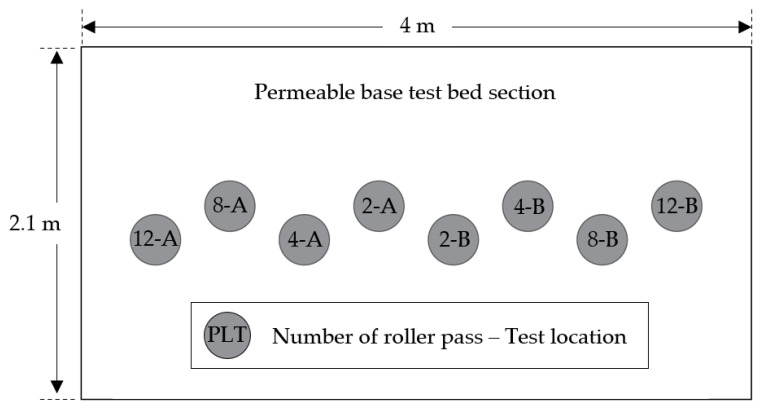
Layout of test locations for PLT [38].

**Figure 7 materials-14-01199-f007:**
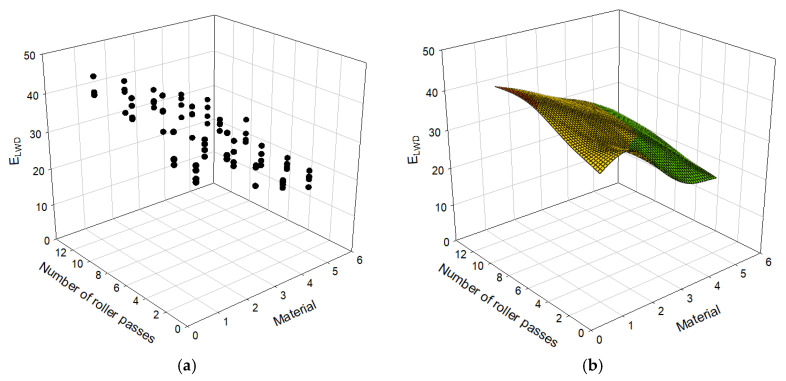
Relationship between number of roller passes and E_LWD_: (**a**) scatter plot, (**b**) surface plot.

**Figure 8 materials-14-01199-f008:**
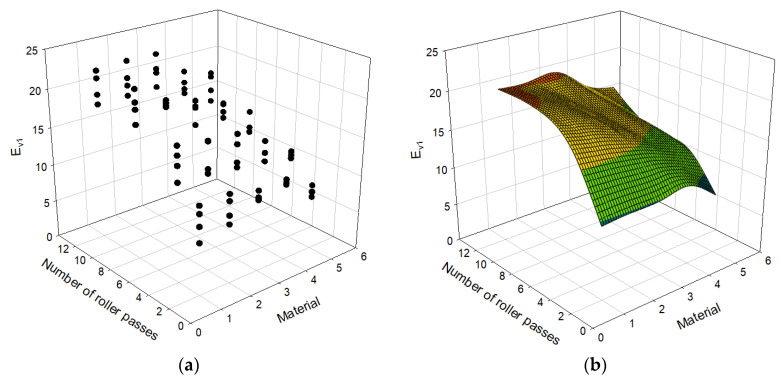
Relationship between number of roller passes and E_v1_: (**a**) scatter plot; (**b**) surface plot.

**Figure 9 materials-14-01199-f009:**
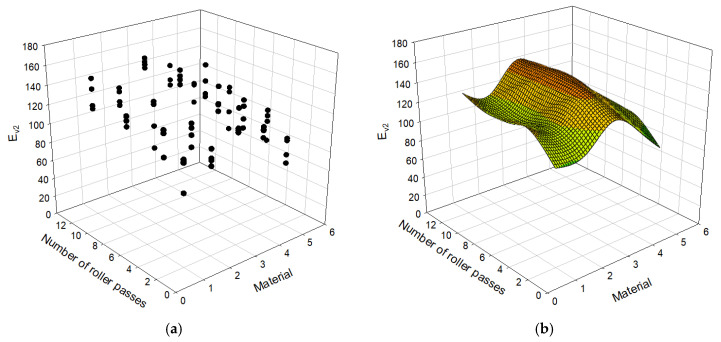
Relationship between the number of roller passes and E_v2_: (**a**) scatter plot, (**b**) surface plot.

**Figure 10 materials-14-01199-f010:**
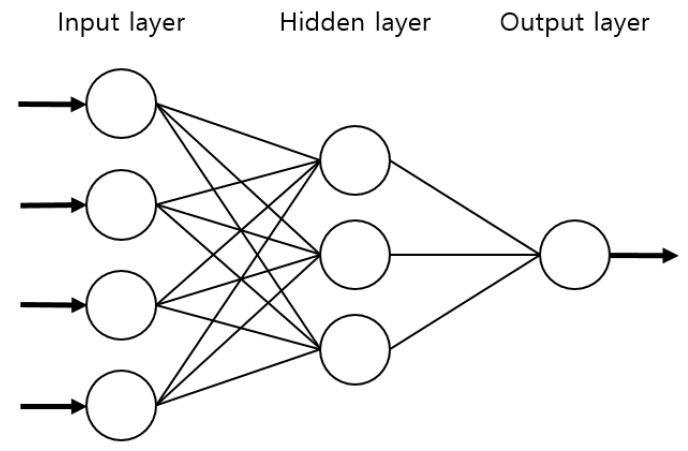
Artificial Neural Network (ANN) structure.

**Figure 11 materials-14-01199-f011:**
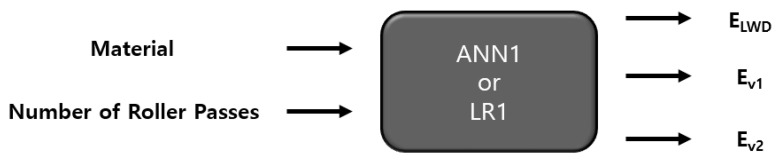
Features and labels for ANN1 and LR1 models.

**Figure 12 materials-14-01199-f012:**
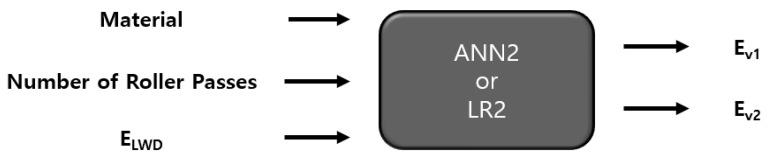
Features and labels for ANN2 and Linear Regression 2 (LR2) models.

**Figure 13 materials-14-01199-f013:**
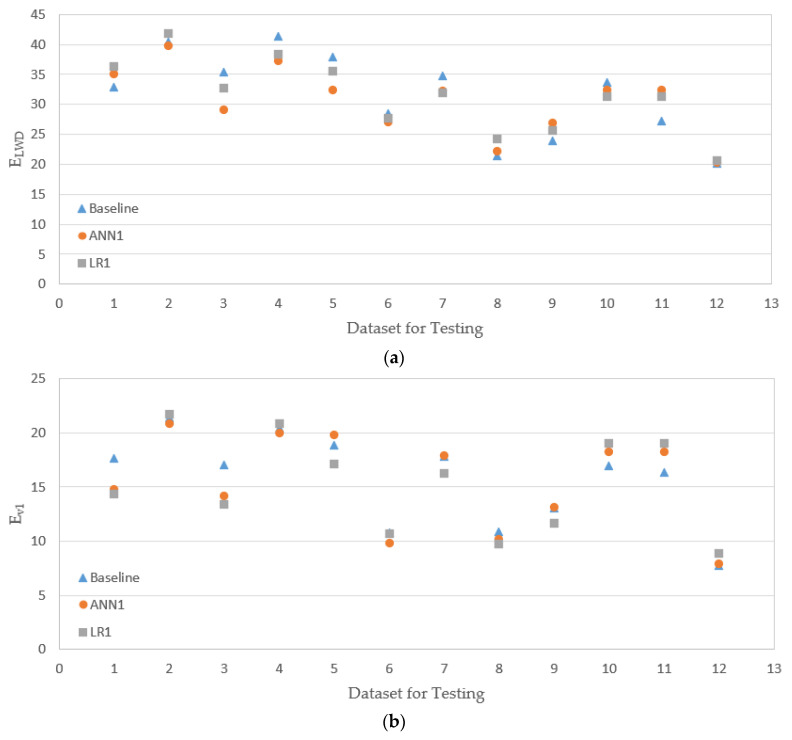
Testing results 1 (results of ANN are based on the model ANN1, and results of LN are based on the model LN1). (**a**) Testing results from E_LWD_; (**b**) testing results from E_v1_; (**c**) testing results from E_v2_.

**Figure 14 materials-14-01199-f014:**
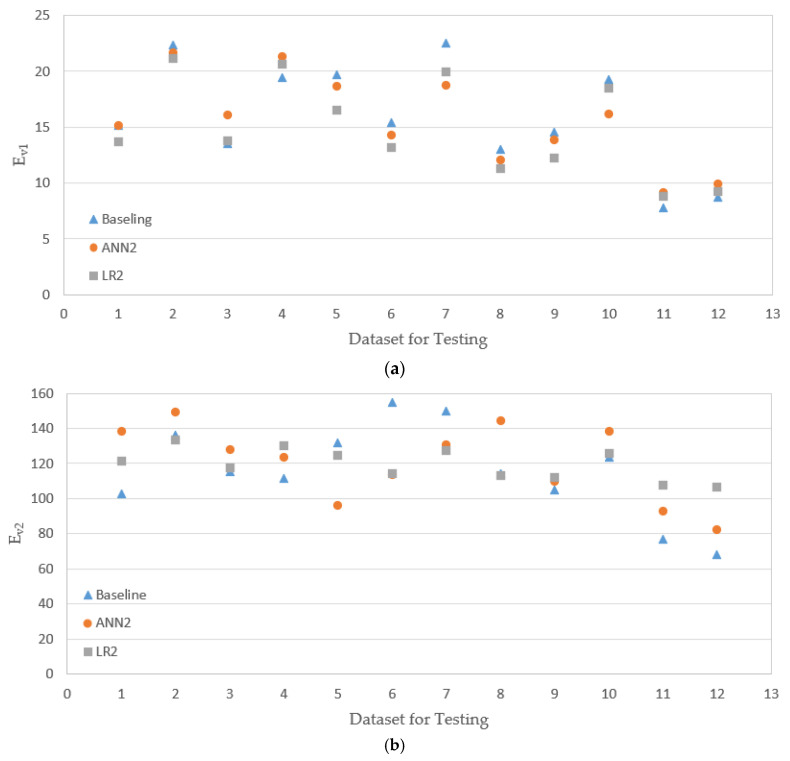
Testing results 2 (results of ANN are based on the model ANN2, and results of LN are based on the model LN2). (**a**) Testing results from E_v1_; (**b**) testing results from E_v2_.

**Table 1 materials-14-01199-t001:** Basic information of the test materials [38].

Test Material	Lithology	Material Composition by Volume			C_u_ ^1^	C_c_ ^2^	Specific Gravity	Abrasion Loss ^3^ (%)
		D40	D25	D13				
D40	Rhyolite	100%	-	-	2.88	1.19	2.67–2.75	12.8
D40 + D25		50%	50%	-	2.99	1.08		9.8
D25		-	100%	-	2.48	1.02		10.3
D25 + D13		-	50%	50%	2.84	1.16		11.2
D13		-	-	100%	2.79	1.16		12.3

^1^ Coefficient of uniformity; ^2^ Coefficient of curvature; ^3^ Loss of the mass of aggregates after the Los Angeles test as a percentage of the original mass of aggregates calculated by ASTM C131 [40].

**Table 2 materials-14-01199-t002:** Test program.

Material	Lift	Number of Roller Passes	Modulus Evaluation
D40	First (30 cm) Second (30 cm)	2 4 8 12	PLT LWD
D40 + D25			
D25			
D25 + D13			
D13			

**Table 3 materials-14-01199-t003:** Dataset.

Dataset No.	Material	Number of Roller Passes	Location				E_LWD_ (MPa)	E_v1_ (MPa)	E_v2_ (MPa)
			First Lift	Second Lift	Area A	Area B			
1	1	2	O		O		32.95	6.92	77.26
2			O			O	34.16	9.03	107.08
3				O	O		30.07	10.62	107.81
4				O		O	30.91	11.68	110.32
5		4	O		O		32.94	17.66	129.37
6			O			O	31.67	15.19	102.47
7				O	O		33.10	16.45	129.01
8				O		O	39.64	13.09	125.98
9		8	O		O		41.39	17.77	125.54
10			O			O	38.47	20.53	124.90
11				O	O		43.27	22.18	114.55
12				O		O	38.09	19.65	120.48
13		12	O		O		44.52	18.07	115.87
14			O			O	39.73	19.32	119.19
15				O	O		40.47	21.40	147.15
16				O		O	39.88	22.35	136.34
17	2	2	O		O		39.48	11.73	111.32
18			O			O	34.11	8.96	99.81
19				O	O		33.21	10.82	101.90
20				O		O	33.16	7.82	94.12
21		4	O		O		32.87	16.97	122.44
22			O			O	31.20	12.91	103.39
23				O	O		35.46	17.01	127.12
24				O		O	34.31	13.49	115.38
25		8	O		O		37.96	18.86	83.60
26			O			O	32.59	19.73	131.75
27				O	O		41.82	19.38	128.94
28				O		O	37.75	19.07	106.75
29		12	O		O		41.33	20.42	129.75
30			O			O	39.10	19.45	111.28
31				O	O		38.50	22.63	125.74
32				O		O	32.94	18.13	115.32
33	3	2	O		O		23.81	9.88	142.27
34			O			O	28.52	10.76	118.15
35				O	O		29.05	9.70	120.44
36				O		O	28.80	9.53	122.32
37		4	O		O		27.26	13.05	137.16
38			O			O	26.24	12.39	137.85
39				O	O		32.73	15.38	154.96
40				O		O	29.93	16.69	130.55
41		8	O		O		34.92	17.68	149.92
42			O			O	28.69	15.43	141.11
43				O	O		36.87	18.59	155.86
44				O		O	34.84	17.85	146.05
45		12	O		O		36.97	22.57	149.79
46			O			O	33.49	20.64	143.28
47				O	O		33.98	20.11	146.78
48				O		O	32.18	18.24	153.81
49	4	2	O		O		22.59	10.17	112.55
50			O			O	21.43	10.87	103.98
51				O	O		20.57	10.39	115.08
52				O		O	22.02	10.38	111.43
53		4	O		O		25.12	13.03	133.66
54			O			O	26.91	11.91	127.44
55				O	O		23.89	13.02	114.34
56				O		O	29.07	14.58	105.12
57		8	O		O		28.18	16.05	154.04
58			O			O	29.95	15.24	137.41
59				O	O		30.21	17.17	124.13
60				O		O	31.26	17.06	121.03
61		12	O		O		33.66	16.94	117.71
62			O			O	30.72	19.30	123.29
63				O	O		32.56	17.82	138.74
64				O		O	27.24	16.35	138.90
65	5	2	O		O		20.19	7.77	77.07
66			O			O	22.51	8.72	68.08
67				O	O		18.03	7.13	94.38
68				O		O	20.89	7.87	92.15
69		4	O		O		20.28	11.92	115.13
70			O			O	20.92	12.00	109.36
71				O	O		23.36	11.47	103.47
72				O		O	21.74	11.03	83.25
73		8	O		O		25.11	14.92	96.55
74			O			O	28.98	12.81	77.88
75				O	O		22.71	12.20	123.13
76				O		O	23.40	12.15	118.23
77		12	O		O		27.90	15.70	90.83
78			O			O	23.23	18.10	90.66
79				O	O		30.14	17.59	111.86
80				O		O	25.37	14.22	110.25

”O” indicates the number of lift and planar location of modulus measurement.

**Table 4 materials-14-01199-t004:** Root mean square error (RMSE) results for ANN1 and LR1.

Modulus	LR1		ANN1	
	Training	Testing	Training	Testing
E_LWD_	2.76	2.53	2.84	3.40
E_v1_	2.00	1.95	1.45	1.42
E_v2_	18.16	17.71	10.84	17.39

**Table 5 materials-14-01199-t005:** RMSE results for ANN2 and LR2.

Modulus	LR2		ANN2	
	Training	Testing	Training	Testing
E_v1_	1.96	1.77	1.49	1.86
E_v2_	17.55	21.23	9.33	23.61

## Data Availability

Data sharing is not applicable to this article.

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
