# Peer review of "Prediction Models Based on Regression and Artificial Neural Network for Moduli of Layers Constituted by Open-Graded Aggregates"

_materials, 2021, doi:10.3390/ma14051199_

Round 1
Reviewer 1 Report
The paper is an interesting work on evaluation of modulus (both static and dynamic) of unbound aggregate mixtures for road base layers. The authors studied models to predict these moduli with a linear regression and/or an Artificial Neural Network. It is not clear if the authors conducted in separate way both the analysis or if these are connected with each other. Please explain this concept in the text because it is not clear.
I suggest the authors to modify the title of the paper in “Prediction Models Based on Regression and Artificial Neural Network for moduli of Layers constituted by Open Graded Aggregates”, because the topic of the paper is the prediction models.
I suggest the authors to check the English.
Abstract (first six lines, from line 14 to line 19) and first section (“1. Introduction”, from line 36 to line 47) are not clear to me. Please consider to re-write these sentences.
Also, subsection “4.2. Modulus results depending on the number of roller passes” (page 10) is not clear. Please check and re-write this part of the paper. For example, the sentence at page 10, lines 176 and 177 “This trend shown by Ev2 is because the large load applied in the first loading cycle compresses the test material to increase the density.” is not English.
Abstract, line 30: I suggest the authors to clarify the concept “number of compactions” indicating method used to compact the unbound aggregates mixture.
Page 1, line 41: what are “low-impact development (LID) techniques”? Please comment.
Page 1, lines 42 and 43: the authors wrote “Permeable pavements are one of the elements of LID technology and are used in several countries worldwide” and cited only one paper published by USA researchers (Chicago). Why the authors wrote “worldwide”? Permeable pavements are used only in Chicago? Please insert other citations to justify the use of the adjective “worldwide”.
Page 3, Figure 3: what is the difference between graphs “a” and “b”? I suggest the authors to insert the letters “a” and “b” in the caption to explain to the reader what the two graphs are.
Page 3, line 80: What is the meaning of the adverb “here”? I suggest the authors to remove this adverb.
I suggest the authors to enlarge the literature review on the “field modulus evaluating devices”.
Page 5, Figure 6: I suggest the authors to insert the letters “a” and “b” in the caption to link the two figures with the text in the caption (as you wrote in your previous work cited – Choi, 2020, i.e. “Conceptual illustration of pavement base materials; (a) open-graded aggregates (OGAs); (b) dense-graded aggregates.”).
Page 5, Figure 7: I suggest the authors to insert the letters “a”, “b” and “c” in the caption to link the three different aggregates with the names (as you wrote in your previous work cited – Choi, 2020, i.e. “The maximum particle sizes are: (a) 40 mm (D40); (b) 25 mm (D25); (c) 13 mm (D13).”).
Pages 5 and 6, Table 1: what is “Abrasion rate”? Please comment/describe in the text. I suggest the authors to comment data in Table 1 and Figure 8.
Page 6, Table 2: what is “lift”? Please describe/link the lift in the text of the paper.
Page 7, Figure 9 need to be split in Figures “9a” and “9b” (as you wrote in your previous work cited – Choi, 2018, i.e. “Test bed layout: (a) Cross-sectional view; (b) Plan view.”).
Page 7, Figure 10: the layout of test locations is repeated for each test bed layout (D40, D40+D25, D25, D25+D13 and D13)? It is not clear to me. I suggest the authors to describe it in the text.
Page 8, lines 163 and 164: I suggest the authors to insert a citation of Table 3 immediately after terms “Area A column” and “Area B column” in the text.
Pages 8, 9 and 10, Table 3: I suggest the authors to insert dotted lines in Table 3 to a better reading of it.
Page 8, Table 3: the explanation of what are materials 1, 2, 3, 4 and 5 is missing in the text. I suggest the authors to insert in the text what are the meaning of the numbers from 1 to 5.
Pages 10 and 11, Figures 11a, 11b and 11c: these Figures are not readable and not comprehensible. It is impossible to read the titles of the axes. Also, the points in the graph are not well-defined. I suggest the authors to insert adequate pictures. Moreover, these figures are not sufficiently cited in the text.
I suggest the authors to comment data presented in Table 8 and in Figures 11a, 11b and 11c.
Page 11, line 189. I suggest the authors to subdivide the subsection 5.1 in two: one for the ANN model and one for linear regression.
Page 12, Figures 12 and 13: these Figures are not insert adequately in the text. The reader can not understand in the text where the authors described these figures.
Page 12, lines 221 and 222: the authors wrote that M1 is the material and N2 is the vibration roller compaction frequency. What is the range of values M1 and N2 considered for the equations from 1 to 5? I suggest the authors to show these values in a Table or in the text.
Why the authors did not consider other linear relations between static and dynamic moduli? The Institute for Transport Sciences of Karlsruhe in Gemany, for example, in the 1995 elaborated the “Baksay” equation, i.e. Ed [MPa] = (0.52∙Es)+9.1 . What think the authors about other relations present in the literature?
In the paper the authors consider the number of roller passes, but in the regression analysis the authors use the vibration roller compaction frequency. Why? Please comment this.
Moreover, how the authors can correlate the energy of compaction (of the roller) with other type of energy of compaction established in the standards or in laboratory test protocols (e.g. Proctor compaction)? Please comment.
Page 12, line 225. I suggest the authors to subdivide the subsection 5.2 in two: one for the ANN model and one for linear regression.
Page 12, lines 226 and 227: the authors wrote that split the sets of data for training, verification and testing. How did the authors do this? What are the rules followed by the authors to split the data?
Page 13, lines 233 and 234: the authors wrote that “the linear regression and ANN prediction results showed no significant differences”. With respect to what there are no significant differences? The sentence is not clear to me.
Pages 13 and 14, Figures 14 and 15: these figures are not indicated in a correct way. Figure 14 is constituted by three figures, meanwhile Figure 15 by two figures. Why did the authors not insert letters that distinguish each of Figures 14 and 15? The captions used are not appropriate. The text in subsection 5.2. is not clear to me. I suggest the authors to insert the letters to distinguish adequately the figures at pages 13 and 14.
Page 15, line 241: What are “A1” and “A2”. In the text are not described. Please insert the description of these.
Page 15, section 6 “Conclusion”. Generally, is not possible to insert in a paper the conclusion in a bulleted list. A scientific paper is not a summary. Please delete the bulleted list and re-write the conclusions. Why did the authors not insert any conclusion with respect the modulus measured with the Light Falling Weight Deflectometer (ELWD)?
References. 20 references are too few. On 20 cited paper, 4 are wrote by the authors (20% percent): it could be too much. I suggest the authors to enlarge the section references with more works. Moreover, I suggest the authors to re-write the references in an alphabetical order.
To help the authors, I show in the following some papers that can help they to enlarge the section references:
- Adam, C.; Adam, D.; Kopf, F.; Paulmichl, I. 2009. Computational validation of static and dynamic plate load testing. Acta Geotechnica, 4(1): 35-55.
- Fleming, P.R.; Frost, M.W.; Lambert, J.P. 2009. Lightweight Deflectometers for quality assurance in road construction. Proceedings of the 8th International Conference on Bearing Capacity of Roads, Railways and Airfields, 809-818.
- Kim, J.R.; Kang, H.B.; Kim, D.; Park, D.S.; Kim, W.J. 2007. Evaluation of in situ modulus of compacted subgrades using portable falling weight deflectometer and plate-bearing load test. Journal of Materials in Civil Engineering, 19(6): 492-499.
- Loizos, A.; Boukovalas, G.; Karlaftis, A. 2003. Dynamic stiffness modulus for pavement subgrade evaluation. Journal of Transportation Engineering, 129(4): 434-443.
- Pasetto, M.; Pasquini, E.; Giacomello. G-; Baliello A. 2018. Proposal of correlations between different soil bearing capacity parameters based on extesive test campaigns. Proceedings of the 4th International Conference on Traffic and Transport Engineering, 577-583
- Patel, M.A.; Patel, H.S. 2013. Laboratory assessment to correlate strength parameter from physical properties of subgrade, Procedia engineering, 51: 200-209.
- Tompai, Z. 2008. Conversion between static and dynamic load bearing capacity moduli and introduction of dynamic target values. Periodica polytechnica Civil Engineering, 52(2): 97-102.
- Wyroslak, M. 2017. Establishing relationships between parameters of the controlled compaction soil by using various in-situ tests. Materials Science and Engineering, 245:1-8.
Author Response
Authors deeply appreciate for the excellent comments of reviewers. Vast majority of the text wer rewritten and new plots were added following reviewers’ comment. The details are following.
Reviewer 1
- The paper is an interesting work on evaluation of modulus (both static and dynamic) of unbound aggregate mixtures for road base layers. The authors studied models to predict these moduli with a linear regression and/or an Artificial Neural Network. It is not clear if the authors conducted in separate way both the analysis or if these are connected with each other. Please explain this concept in the text because it is not clear.
Title of a sub-section has been changed, and the separate use of LR and ANN is clarified. “5.1. Linear regression and ANN models - It was attempted to evaluate the moduli of open-graded aggregates based on either linear regression or ANN, in order to see whether the moduli can be predicted well from aggregate and compaction information. MATLAB was used to deploy the ANN.”
- I suggest the authors to modify the title of the paper in “Prediction Models Based on Regression and Artificial Neural Network for moduli of Layers constituted by Open Graded Aggregates”, because the topic of the paper is the prediction models.
The title of the paper has been changed to “Prediction Models Based on Regression and Artificial Neural Network for moduli of Layers constituted by Open-Graded Aggregates”.
- I suggest the authors to check the English.
Most of the manuscript has been rewritten based on the reviewers’ comments. The manuscript has been checked thoroughly.
- Abstract (first six lines, from line 14 to line 19) and first section (“1. Introduction”, from line 36 to line 47) are not clear to me. Please consider to re-write these sentences.
Corresponding part of abstract has been re-written as “The impermeable cover in urban area has been growing due to rapid urbanization, which prevents stormwater naturally infiltrated into the ground. There is higher chance of flooding in urban area covered with conventional concretes and asphalts. The permeable pavement is one of Low Impact Development technologies that can reduce surface runoff and water pollution by allowing stormwater into pavement systems”
The first section has been re-written as “The rainfall pattern changes globally due to climate change, and therefore the rainfall can be much more intense than in the past. In addition, the impermeable cover in urban area has been growing due to rapid urbanization [1, 2]. Owing to climate change and urbanization, the cities suffer from heavier flood damages nowadays. Low Impact Development (LID) is a design philosophy that can remedy these issues, often using surface-infiltrating stormwater facilities, and is being widely implemented worldwide [1-5]. The permeable pavements are one of the most popular LID technologies which can especially be well applied in urban area [6-12]. Figure 1 shows typical sections of permeable pavements. Permeable pavements bring other benefits such as reducing the heat island phenomenon, leveling down traffic noise, and providing better skid resistance [9].”
- Also, subsection “4.2. Modulus results depending on the number of roller passes” (page 10) is not clear. Please check and re-write this part of the paper. For example, the sentence at page 10, lines 176 and 177 “This trend shown by Ev2 is because the large load applied in the first loading cycle compresses the test material to increase the density.” is not English.
The section has been re-written as “The moduli Ev1, Ev2, and ELWD are plotted with respect to the number of roller passes and material type in Figure 7, Figure 8, and Figure 9 with the data presented in Table 3. Both the scatter plot and surface plot, which consists of the average values of the data points of the scatter plot, is shown in those figures to visualize data better. The Ev1 and Ev2 values steeply increase until the number of roller passes reaches 4. After 4 passes, Ev1 continues to increase with increasing number of roller passes with less steep rate, which was not the case for Ev2 and ELWD. During the first loading, the specimens are compressed and densified (or compacted), and therefore the second compression curve (or Ev2) is less sensitive to the number of roller passes. The first loading of plate load test also involves significant shear displacement, which does not happen during the second static loading or dynamic loading; Ev1 is more sensitive to the number of roller passes than Ev2 and ELWD overall. All three moduli Ev1, Ev2, and ELWD values nearly settle after the number of roller passes becomes 8. The number of roller passes 4, which is typically used in practice, may be an efficient number, but one may consider applying the number of roller passes 8 to achieve better stiffness for open-graded aggregates.”
- Abstract, line 30: I suggest the authors to clarify the concept “number of compactions” indicating method used to compact the unbound aggregates mixture.
It is “the number of roller passes” with field compactor. The term has been replaced throughout the manuscript.
- Page 1, line 41: what are “low-impact development (LID) techniques”? Please comment.
It is added and explained in the section. “Low Impact Development (LID) is a design philosophy that can remedy these issues, often using surface-infiltrating stormwater facilities, and is being widely implemented worldwide.”
- Page 1, lines 42 and 43: the authors wrote “Permeable pavements are one of the elements of LID technology and are used in several countries worldwide” and cited only one paper published by USA researchers (Chicago). Why the authors wrote “worldwide”? Permeable pavements are used only in Chicago? Please insert other citations to justify the use of the adjective “worldwide”
References are added and cited.
- Page 3, Figure 3: what is the difference between graphs “a” and “b”? I suggest the authors to insert the letters “a” and “b” in the caption to explain to the reader what the two graphs are.
The authors have determined to remove figure 3 for the consistence and following reviewers’ comments.
- Page 3, line 80: What is the meaning of the adverb “here”? I suggest the authors to remove this adverb.
It has been removed.
- I suggest the authors to enlarge the literature review on the “field modulus evaluating devices”.
The reviews on field modulus evaluating devices are added.
- Page 5, Figure 6: I suggest the authors to insert the letters “a” and “b” in the caption to link the two figures with the text in the caption (as you wrote in your previous work cited – Choi, 2020, i.e. “Conceptual illustration of pavement base materials; (a) open-graded aggregates (OGAs); (b) dense-graded aggregates.”).
The authors removed the figure for improve the condensity of the manuscript because it was not essential part following reviewers’ comments.
- Page 5, Figure 7: I suggest the authors to insert the letters “a”, “b” and “c” in the caption to link the three different aggregates with the names (as you wrote in your previous work cited – Choi, 2020, i.e. “The maximum particle sizes are: (a) 40 mm (D40); (b) 25 mm (D25); (c) 13 mm (D13).”).
The authors removed the figure for improve the condensity of the manuscript because it was not essential part following reviewers’ comments.
- Pages 5 and 6, Table 1: what is “Abrasion rate”? Please comment/describe in the text. I suggest the authors to comment data in Table 1 and Figure 8.
The term “abrasion rate” has been modified to “abrasion loss” and the description for it is added as a foot note. In addition, a comment on Figure 8 has been added.
- Page 6, Table 2: what is “lift”? Please describe/link the lift in the text of the paper.
Description has been added
- Page 7, Figure 9 need to be split in Figures “9a” and “9b” (as you wrote in your previous work cited – Choi, 2018, i.e. “Test bed layout: (a) Cross-sectional view; (b) Plan view.”).
The figures have been split to (a) and (b).
- Page 7, Figure 10: the layout of test locations is repeated for each test bed layout (D40, D40+D25, D25, D25+D13 and D13)? It is not clear to me. I suggest the authors to describe it in the text.
Description about it has been added.
- Page 8, lines 163 and 164: I suggest the authors to insert a citation of Table 3 immediately after terms “Area A column” and “Area B column” in the text.
The citation of the table has been added
- Pages 8, 9 and 10, Table 3: I suggest the authors to insert dotted lines in Table 3 to a better reading of it.
The authors have improved the readability of the Table and condensed it.
- Page 8, Table 3: the explanation of what are materials 1, 2, 3, 4 and 5 is missing in the text. I suggest the authors to insert in the text what are the meaning of the numbers from 1 to 5.
The authors have added an explanation of the number 1 to 5 in the text before Table 3 appears.
- Pages 10 and 11, Figures 11a, 11b and 11c: these Figures are not readable and not comprehensible. It is impossible to read the titles of the axes. Also, the points in the graph are not well-defined. I suggest the authors to insert adequate pictures. Moreover, these figures are not sufficiently cited in the text.
The figures are newly drawn and discussions are also rewritten following reviewers’ comments.
- I suggest the authors to comment data presented in Table 8 and in Figures 11a, 11b and 11c.
Comments have been added to the figures mentioned.
- Page 11, line 189. I suggest the authors to subdivide the subsection 5.1 in two: one for the ANN model and one for linear regression.
The authors have subdivided the section 5.1 as suggested.
- Page 12, Figures 12 and 13: these Figures are not insert adequately in the text. The reader can not understand in the text where the authors described these figures.
Citations of the figures have been added to help readers to identify which text explains those figures and vice versa.
- Page 12, lines 221 and 222: the authors wrote that M1 is the material and N2 is the vibration roller compaction frequency. What is the range of values M1 and N2 considered for the equations from 1 to 5? I suggest the authors to show these values in a Table or in the text.
The ranges of values M1 and N2 had been added. Please note that the term M1 and N2 has changed to Nmat and Nrp to better reflect its meaning.
- Why the authors did not consider other linear relations between static and dynamic moduli? The Institute for Transport Sciences of Karlsruhe in Gemany, for example, in the 1995 elaborated the “Baksay” equation, i.e. Ed [MPa] = (0.52∙Es)+9.1 . What think the authors about other relations present in the literature?
Because the previous research focused on conventional road embankment materials to establish the relationship, and other studies also did so, the authors thought that the relationship might not be applicable to the materials addressed in this study. However, we included a few literature reviews on the previous research that investigated the relationship between various modulus measures focusing on the conventional materials, such as Kim D (2011) and Abu-Farsakh (2004)
- In the paper the authors consider the number of roller passes, but in the regression analysis the authors use the vibration roller compaction frequency. Why? Please comment this.
We have corrected “the frequency” to “the number of roller passes” throughout the manuscript. We apologize for the errors and confusion.
- Moreover, how the authors can correlate the energy of compaction (of the roller) with other type of energy of compaction established in the standards or in laboratory test protocols (e.g. Proctor compaction)? Please comment.
The way compaction energy is applied in the field (kneading action) is very different from the way in the lab (impact or vibration). In addition, it is very hard to evaluate the magnitude of energy in the field as it quite is sensitive to the operation and ground conditions. It is explained in the manuscript as following. “In addition, the experimental study conducted in lab environment does not necessary reflect what would happen in the field. The magnitude of compaction energy and the way that the energy is delivered cannot be the same in the field and lab.”
- Page 12, line 225. I suggest the authors to subdivide the subsection 5.2 in two: one for the ANN model and one for linear regression.
The purpose of the section was to compare the model performance of the ANN and linear regression together, the authors would like to describe those models in the same section. Instead, the subsection 5.1 has been separated into two, one for ANN and one for linear regression considering what the comment implies.
- Page 12, lines 226 and 227: the authors wrote that split the sets of data for training, verification and testing. How did the authors do this? What are the rules followed by the authors to split the data?
The ratio typically employed for ANN, was used; given the ratio, data sets were randomly selected. It is added in the paper. “The experimental data had total 80 sets as presented in Table 3, and 70%, 15%, and 15% of data were randomly selected and used for training, validation, and testing (prediction), respectively, with an option implemented in MatLab.”
- Page 13, lines 233 and 234: the authors wrote that “the linear regression and ANN prediction results showed no significant differences”. With respect to what there are no significant differences? The sentence is not clear to me.
The discussion was elaborated and presente the manuscript. “Table 4 presents the RMSE values for training and testing of ANN1 and LR1 (see Figure 13). In case of the model LR1, for all three moduli ELWD, Ev1 and Ev2, the RMSE values for training (fitting) and testing (prediction) are quite close; during fitting and evaluation the LR1 model produces similar level of errors. The model ANN1, on the other hand, for ELWD and Ev2, the RMSE for testing is larger than that for testing. When the testing RMSEs of two different models ANN1 and LR1 are compared, RMSE of ANN1 is higher for ELWD, but RMSE of LR1 is higher for Ev1. One model is not necessarily superior to the other for prediction. The results of the model evaluations of ANN2 and LR2 (see Figure 14) are presented in Table 5. The ANN2 model works better for training (or fitting), but the LR2 model is slightly better at testing (prediction).”
- Pages 13 and 14, Figures 14 and 15: these figures are not indicated in a correct way. Figure 14 is constituted by three figures, meanwhile Figure 15 by two figures. Why did the authors not insert letters that distinguish each of Figures 14 and 15? The captions used are not appropriate. The text in subsection 5.2. is not clear to me. I suggest the authors to insert the letters to distinguish adequately the figures at pages 13 and 14.
Letters to distinguish each of the Figures have been inserted and more detailed descriptions have been added to the titles of the figures for better clarity.
- Page 15, line 241: What are “A1” and “A2”. In the text are not described. Please insert the description of these.
Citations of figures for describing “A1” and “A2” have been added.
- Page 15, section 6 “Conclusion”. Generally, is not possible to insert in a paper the conclusion in a bulleted list. A scientific paper is not a summary. Please delete the bulleted list and re-write the conclusions. Why did the authors not insert any conclusion with respect the modulus measured with the Light Falling Weight Deflectometer (ELWD)?
The conclusion was rewritten and it also discusses ELWD following reviewers’ comment. Another review requested to keep the bullet points that they are kept.
- References. 20 references are too few. On 20 cited paper, 4 are wrote by the authors (20% percent): it could be too much. I suggest the authors to enlarge the section references with more works. Moreover, I suggest the authors to re-write the references in an alphabetical order.
More literature reviews have been carried out and included in the text. Thank you for pointing that out.
Reviewer 2 Report
The authors present a stiffness study of open-graded aggregates (OGA) applied in permeable road pavement base layers. The stiffness is evaluated for 5 types of materials and 4 different degrees of compaction using the plate load test (PLT) and the light weight deflectometer (LWD). Linear regression and artificial neural networks analysis were applied to the results obtained to predict Ev1, Ev2 and Elwd. Prediction results were compared with experimental results.
Dear Authors, please consider the following comments:
General comment:
Most of the information presented (test procedure, results obtained in PLT and LWD and interpretation of these results) has already been published by the authors in Sustainability - MDPI (Assessment of Field Compaction of Aggregate Base Materials for Permeable Pavements Based on Plate Load Tests - 2018, Compaction Quality Monitoring of Open-Graded Aggregates by Light Weight Deflectometer and Soil Stiffness Gauge - 2020). Thus, the information presented in sections 2, 3 and 4 has already been published. In relation to section 5, which presents the use of linear regression and ANN to predict the stiffness of the tested materials, it lacks a more detailed explanation and justification, statistically supported, to effectively prove the validity of Ev1, Ev2 and Elwd predictions
Specific comments:
Abstract and Keywords
- Abstract needs revision. The connection between the introductory part on permeable pavements in urban areas and what is actually presented in the study is not clear. It is also not completely clear that the study focuses on OGAs materials applied in road pavement base layers and the gap or novelty addressed.
- Keywords – suggestion: include "modulus" and remove "low impact development".
- Introduction
- This section should include a broader bibliographic review on permeable layers of road pavements.
- Figure 1 must be presented / explained in the body of the text (all figures must be presented in the text – review for Figures 1, 9, 12 and 13).
- Field modulus evaluating devices
- Page 2 – lines 72-74: include bibliographic references of the referred standards.
- Caption and graphs in Figure 3 are only understood after reading section 3 (field experiment) – improve/review.
- Page 3 - line 80: why start with “Here, Elwd is the …"?
- Information presented in page 4 – lines 94-107 is not clear – review.
- Page 5 – line 110 – “…few studies on OGAs to date…” include citations of the few studies performed.
- Table 1 – units are missing – review.
- Figure 6 and 7 - Include the letters (a), (b), (c) and meaning in caption. Figure 8 – correct citation [29].
- Comment on the materials studied in view of the standards limits presented. Clarify why these 5 materials (and not others) were studied.
- Table 2 can be removed, since it repeats the information presented in the text.
- An area of 4.0m x 2.1m for each type of material is representative and is the spacing between the 24 tests carried out in this area sufficient to guarantee reliable results - not influenced by the tests carried out in the vicinity? Include a comment on these aspects.
- Modulus of open-graded aggregate
- Page 7 – line 161: “80 types of materials” or “80 cases”? (16 cases per type of material)
- Include in the text a brief explanation of the following aspects: what Ev1, Ev2 and Elwd represent? Clarify what the first lift, second lift, area A and area B columns represent.
- Discussion on Ev1, Ev2 and Eldw increase rates should be presented and supported by values clearly visible in graphs. The 3D graphs in Figure 11 are not perceptible. Please consider for better interpretation the use of colours or the preparation of line graphs (already presented in a previous publication of these results) .
- Variation of Elwd results with the number of roller passes should be improved.
- Also consider to include a comment on the probable causes related to results trend deviations (proximity of tests, mixture of aggregates?)
- Linear regression and ANN using matlab
- Support the information presented on ANN and linear regression with citations and references.
- Why not consider an ANN A3 model with M1, N2, Ev1 and Ev2 inputs and Elwd output? The same for linear regression, ie, why was not considered an equation for Elwd as a function of M1, N2, Ev1 and Ev2?
- What specific input information about the tested materials (M1) is used in the AAN models and linear regressions? This aspect must be clearly presented in the text.
- The fit of the obtained models to the real data must be analysed based on the interpretation of statistical tests that allow to evaluate this aspect (for example, in the linear regressions through the R2-value and the standard error of the estimates).
- Page 12 - line 226-227: taking into account that 5 materials were tested and 4 cases of compaction were considered, how were the percentages defined and cases assigned for training, testing and validation? Do the authors mean 22% instead of 12%?
- Page 13 – line 234: Author’s state that the results obtained in ANN and linear regression predictions are not significantly different - What supports this statement? Clarify.
- Figure 15 - why are only 12 cases presented? These are the verification cases?
- Are RMSE values of 17 or 23 acceptable? Support / Justify.
- Clarify whether the models, equations and RMSE values were obtained with all cases (5 types of materials and 4 types of compaction), training representative data or other set of data. This aspect is not clearly presented in the manuscript.
Figures
- Correct numbering of Figures (there are two Figures 12).
6. Conclusions
- Conclusions should be reviewed taking into account the aspects to be improved, especially the ones pointed to section 5.
- Clearly state the scientific gap or novelty addressed with the results obtained.
- Include future work to be developed - for example: in addition to the mechanical characteristics of the OGAs, the characterization of their permeability can support the decision on the best material to use in permeable pavements base layers.
Author Response
Authors deeply appreciate for the excellent comments of reviewers. Vast majority of the text wer rewritten and new plots were added following reviewers’ comment. The details are following.
Reviewer 2
- General comment:
- Most of the information presented (test procedure, results obtained in PLT and LWD and interpretation of these results) has already been published by the authors in Sustainability - MDPI (Assessment of Field Compaction of Aggregate Base Materials for Permeable Pavements Based on Plate Load Tests - 2018, Compaction Quality Monitoring of Open-Graded Aggregates by Light Weight Deflectometer and Soil Stiffness Gauge - 2020). Thus, the information presented in sections 2, 3 and 4 has already been published. In relation to section 5, which presents the use of linear regression and ANN to predict the stiffness of the tested materials, it lacks a more detailed explanation and justification, statistically supported, to effectively prove the validity of Ev1, Ev2 and Elwd predictions
Authors tried and modified to make sections 2, 3, and 4 more concise. In addition, in section 5, detailed discussions were made.
- Specific comments:
Abstract and Keywords
- Abstract needs revision. The connection between the introductory part on permeable pavements in urban areas and what is actually presented in the study is not clear. It is also not completely clear that the study focuses on OGAs materials applied in road pavement base layers and the gap or novelty addressed.
Abstract has been rewritten to make the motivation and novelty clear.
- Keywords – suggestion: include "modulus" and remove "low impact development".
The authors have removed “low impact development” and added “modulus” as suggested.
- Introduction
- This section should include a broader bibliographic review on permeable layers of road pavements.
Literature reviews have been expanded and included in the manuscript.
- Figure 1 must be presented / explained in the body of the text (all figures must be presented in the text – review for Figures 1, 9, 12 and 13).
The authors have included explanations or citations of Figures 1, 9, 12, and 13.
- Field modulus evaluating devices
- Page 2 – lines 72-74: include bibliographic references of the referred standards.
The References for those standards have been added.
- Caption and graphs in Figure 3 are only understood after reading section 3 (field experiment) – improve/review.
The figures are simplified to one figure to help readers understand the concept of the test better and to get rid of complexity, and a corresponding additional explanation is added in those sections.
- Page 3 - line 80: why start with “Here, Elwd is the …"?
Some corrections and improvements in the text have been included to explain LWD and ELWD better.
- Information presented in page 4 – lines 94-107 is not clear – review.
We have improved the sentences.
- Page 5 – line 110 – “…few studies on OGAs to date…” include citations of the few studies performed.
There were minor errors in those sentences. As far as the author’s knowledge, there is no research about the modulus-based compaction quality control of OGAs except for the previous research conducted by the authors, and most studies regarding it have been focused on conventional road base materials. The sentence has been changed to express this aspect in the manuscript.
- Table 1 – units are missing – review.
The units of all the values in Table 1 are checked and presented properly.
- Figure 6 and 7 - Include the letters (a), (b), (c) and meaning in caption. Figure 8 – correct citation [29].
Descriptions for (a), (b), and (c) has been added in Figure 7, and citation has been corrected in Figure 8.
- Comment on the materials studied in view of the standards limits presented. Clarify why these 5 materials (and not others) were studied.
The particle sizes of those 5 materials approximately lie in a commonly used particle size range of the base materials of permeable pavements. This reason has been included in the manuscript.
- Table 2 can be removed, since it repeats the information presented in the text.
The authors tried to make the manuscript concise and neat overall. In the case of Table 2, there was a request of another reviewer and the authors keep this table in the manuscript.
- An area of 4.0m x 2.1m for each type of material is representative and is the spacing between the 24 tests carried out in this area sufficient to guarantee reliable results - not influenced by the tests carried out in the vicinity? Include a comment on these aspects.
LWD is non-destructive test and therefore this can be applied near the test of its own kind or other type. PLT, however, involved high deformation and therefore it is important to make sure they do not interfere each other. The size of plate of PLT test is 30 cm, and each of PLT is at lease 30 cm, apart from another. Considering, the shape of pressure bulb, and the dimension, and also highly localized shape of deformation, each PLT would not affect another.
- Modulus of open-graded aggregate
- Page 7 – line 161: “80 types of materials” or “80 cases”? (16 cases per type of material)
Thank you for pointing that out. The sentence has been improved.
- Include in the text a brief explanation of the following aspects: what Ev1, Ev2 and Elwd represent? Clarify what the first lift, second lift, area A and area B columns represent.
Explanations for Ev1, Ev2, Elwd have been improved or added in section 2. In addition, descriptions for area A and B have been improved and incorporated in the manuscript.
- Discussion on Ev1, Ev2 and Eldw increase rates should be presented and supported by values clearly visible in graphs. The 3D graphs in Figure 11 are not perceptible. Please consider for better interpretation the use of colours or the preparation of line graphs (already presented in a previous publication of these results)
The picture has been modified.
- Variation of Elwd results with the number of roller passes should be improved.
Following the comments, the moduli versus the material type and number of roller passes are newly drawn with rewritten texts in section 4.2.
- Also consider to include a comment on the probable causes related to results trend deviations (proximity of tests, mixture of aggregates?)
The comments on field compaction and differences of measuring devices in the field have been elaborated in the manuscript. “In addition, the experimental study conducted in lab environment does not necessary reflect what would happen in the field. The magnitude of compaction energy and the way that the energy is delivered cannot be the same in the field and lab. In this study, the moduli of open-graded aggregates of various sizes under various compaction energies are investigated based on PLT (Plate Load Test) LWD (Lightweight Defectometer). There has been very few experimental study on the stiffness of open-grade aggregates in field condition.” “PLT costs more much more time and efforts to conduct that LWD, and replacing PLT by LWD would make quality assessment procedure simpler. The intention of models ANN2 and LR2 are, to predict the results of PLT based on those of LWD.” “During the first loading, the specimens are compressed and densified (or compacted), and therefore the second compression curve (or Ev2) is less sensitive to the number of roller passes. The first loading of plate load test also involves significant shear displacement, which does not happen during the second static loading or dynamic loading; Ev1 is more sensitive to the number of roller passes than Ev2 and ELWD overall.”
- Linear regression and ANN using matlab
- Support the information presented on ANN and linear regression with citations and references.
Reference and citations have been added [41-44, 47].
- Why not consider an ANN A3 model with M1, N2, Ev1 and Ev2 inputs and Elwd output? The same for linear regression, ie, why was not considered an equation for Elwd as a function of M1, N2, Ev1 and Ev2?
PLT costs more much more time and efforts to conduct that LWD, and replacing PLT by LWD would make quality assessment procedure simpler. The intention of models ANN2 and LR2 are, to predict the results of PLT based on those of LWD. While the models ANN1 and LR1 employs only the material type and number of roller passes as inputs, the models ANN2 and LR2 has additionally ELWD as an input. It is elaborated and added in the manuscript.
- What specific input information about the tested materials (M1) is used in the AAN models and linear regressions? This aspect must be clearly presented in the text.
Information about the values of M1 has been added in the manuscript. Note that the term M1 has been changed to Nmat.
- The fit of the obtained models to the real data must be analysed based on the interpretation of statistical tests that allow to evaluate this aspect (for example, in the linear regressions through the R2-value and the standard error of the estimates).
The authors have included RMSE analysis in Tables 4 and 5 and elaborated them in the manuscript to reflect the reviewer’s comment.
- Page 12 - line 226-227: taking into account that 5 materials were tested and 4 cases of compaction were considered, how were the percentages defined and cases assigned for training, testing and validation? Do the authors mean 22% instead of 12%?
The explanation about the reviewer’s comment has benn included in section 5.2 “The experimental data had total 80 sets as presented in Table 3, and 70%, 15%, and 15% of data were randomly selected and used for training, validation, and testing (prediction), respectively, with an option implemented in MatLab.”
- Page 13 – line 234: Author’s state that the results obtained in ANN and linear regression predictions are not significantly different - What supports this statement? Clarify.
Authors elaborated it and presented in the paper. “Table 4 presents the RMSE values for training and testing of ANN1 and LR1 (see Figure 13). In case of the model LR1, for all three moduli ELWD, Ev1 and Ev2, the RMSE values for training (fitting) and testing (prediction) are quite close; during fitting and evaluation the LR1 model produces similar level of errors. The model ANN1, on the other hand, for ELWD and Ev2, the RMSE for testing is larger than that for testing. When the testing RMSEs of two different models ANN1 and LR1 are compared, RMSE of ANN1 is higher for ELWD, but RMSE of LR1 is higher for Ev1. One model is not necessarily superior to the other for prediction. The results of the model evaluations of ANN2 and LR2 (see Figure 14) are presented in Table 5. The ANN2 model works better for training (or fitting), but the LR2 model is slightly better at testing (prediction).”
- Figure 15 - why are only 12 cases presented? These are the verification cases?
From 80 sets of data, 56 (70%) of them were used as model training, 12 (15%) of them were used as model validation which is a step for improving the model, and the other 12 (15%) were used as a prediction, whose results are presented in Figure 15 and 16. The authors have made some changes in the text for improving clarity.
- Are RMSE values of 17 or 23 acceptable? Support / Justify.
RMSEs of Ev2 are high because the values of Ev2 are larger than the values of Ev1 and ELWD. There is no criterion or threshold, but it seems the presented models reasonable well follow the baselines as presented the figures where the comparions of the predictions and the baselines are made.
- Clarify whether the models, equations and RMSE values were obtained with all cases (5 types of materials and 4 types of compaction), training representative data or other set of data. This aspect is not clearly presented in the manuscript.
Section 5 is rearranged and rewritten to present them better following reviewers’ comments.
- Figures
- Correct numbering of Figures (there are two Figures 12).
Numberings of Figures have been corrected.
- Conclusions
- Conclusions should be reviewed taking into account the aspects to be improved, especially the ones pointed to section 5.
The conclusion has been completely rewritten and presented in the manuscript.
- Clearly state the scientific gap or novelty addressed with the results obtained.
The behavior or mechanically properties of dense granular materials have been widely studied worldwird. However, open-graded aggregates has not been studied much; there are very few. This study is one of few studies (if not the only) on OGA especially in field scale. Authors pointed it out in the paper. “In addition, the experimental study conducted in lab environment does not necessary reflect what would happen in the field. The magnitude of compaction energy and the way that the energy is delivered cannot be the same in the field and lab. In this study, the moduli of open-graded aggregates of various sizes under various compaction energies are investigated based on PLT (Plate Load Test) LWD (Lightweight Defectometer). There has been very few experimental study on the stiffness of open-grade aggregates in field condition.”
- Include future work to be developed - for example: in addition to the mechanical characteristics of the OGAs, the characterization of their permeability can support the decision on the best material to use in permeable pavements base layers.
OGA materials are highly porous (40%) and the size of pore is also very large. As such, when the water is applied on compacted OGA, it just immediately drains and the concept or definition of permeability can not apply. It is not laminar flow (basis for Darcy’s law) at all.
Reviewer 3 Report
This is an interesting work. Some comments:
- Abstract, Line 22: Open-Graded Aggregates or Porous Materials is not a new structure. This structure was initially developed for traffic noise reduction purpose decades ago. If the authors aim at the OGA, their mechanical properties have been well investigated already. The coupling mechanical properties of OGA and water have not been studied deeply. Please revise it.
- Line 44: if this paper focuses on the low-traffic roads, please point it out.
- Line 47: special filter course is needed to prevent road surface freezing. Please delete this sentence or revise it.
- Line 51: This is for the conventional dense pavement, please cite a reference about OGA pavement. According to my knowledge, the compaction level of the base layer is not the most important issue that affects the durability of permeable pavement.
- Figure 1: The surface of permeable pavement may have different types. Please also include Figure 1-3 in ''Eisenberg, B., Lindow, K. C., & Smith, D. R. (Eds.). (2015, March). Permeable pavements. American Society of Civil Engineers''. And Figure 1 is not mentioned in the manuscript.
- Line 73: do the authors follow the German standard?
- Please provide some basic information on the surface layer, different surface layer can significantly affect the results.
- Line 125: how the authors decide the three levels of natural dryness?
Author Response
Authors deeply appreciate for the excellent comments of reviewers. Vast majority of the text wer rewritten and new plots were added following reviewers’ comment. The details are following.
Reviewer 3
- Abstract, Line 22: Open-Graded Aggregates or Porous Materials is not a new structure. This structure was initially developed for traffic noise reduction purpose decades ago. If the authors aim at the OGA, their mechanical properties have been well investigated already. The coupling mechanical properties of OGA and water have not been studied deeply. Please revise it.
The behavior or mechanically properties of dense granular materials have been widely studied worldwird. However, open-graded aggregates has not been studied much; there are very few. This study is one of few studies (if not the only) on OGA especially in field scale. Authors pointed it out in the paper. “In addition, the experimental study conducted in lab environment does not necessary reflect what would happen in the field. The magnitude of compaction energy and the way that the energy is delivered cannot be the same in the field and lab. In this study, the moduli of open-graded aggregates of various sizes under various compaction energies are investigated based on PLT (Plate Load Test) LWD (Lightweight Defectometer). There has been very few experimental study on the stiffness of open-grade aggregates in field condition.”
- Line 44: if this paper focuses on the low-traffic roads, please point it out.
There is actually an argument on this. Some say permeable pavements are suitable for only low level traffic loads, while in some area permeable pavements are operated. The introduction section has been rewritten and authors deleted the comment on low-traffic roads.
- Line 47: special filter course is needed to prevent road surface freezing. Please delete this sentence or revise it.
It has been deleted.
- Line 51: This is for the conventional dense pavement, please cite a reference about OGA pavement. According to my knowledge, the compaction level of the base layer is not the most important issue that affects the durability of permeable pavement.
The authors agree with the reviewer’s comment. The compaction can be one of the very important processes that affect the durability of permeable pavements, but not the most important one. The sentence has been corrected reflecting the reviewer’s comment.
- Figure 1: The surface of permeable pavement may have different types. Please also include Figure 1-3 in ''Eisenberg, B., Lindow, K. C., & Smith, D. R. (Eds.). (2015, March). Permeable pavements. American Society of Civil Engineers''. And Figure 1 is not mentioned in the manuscript.
The suggested figure has been included in Figure 1 and the citation has been added in the text.
- Line 73: do the authors follow the German standard?
Yes. We have added a comment in the text to clarify it.
- Please provide some basic information on the surface layer, different surface layer can significantly affect the results.
The whole field test was conducted on the base layers constructed with open-graded aggregates without installing surface layers because the main purpose of the test was to evaluate the mechanical performance and modulus of the open-graded aggregate base layers. I hope this might be a sufficient answer to the reviewer’s comment.
- Line 125: how the authors decide the three levels of natural dryness?
The sentence was misleading. We have corrected it to “The aggregates were prepared in an air-dried state with about 0.5% of water contents”
Reviewer 4 Report
Please find attached a PDF file with my comments and suggestions for authors.

Author Response
Authors deeply appreciate for the excellent comments of reviewers. Vast majority of the text wer rewritten and new plots were added following reviewers’ comment. The details are following.
Reviewer 4
- First of all, as general comment, the format of the references cited in the text of the manuscript is not correct, because numbers must be used according to Materials journal template.
Reference and citation format has been corrected to fit the Materials journal template.
- Regarding the introduction section, the state‐of‐art review included here must be improved. The current state‐of‐art review performed is very short and only 4 references have been cited. This is a very low number, which suggest a poor review related to the topic of the manuscript. In addition to this, I cannot see in that section clearly defined the aim and the novelty of the research included in the paper. In my opinion, authors must include a final paragraph in which this aim and novelty would be explicitly explained.
More literature reviews have been included throughout section 1, which focuses on the LID technique, and also throughout section 2, which focuses on permeable pavements, and the part that explains the purpose and novelty of the research have been improved at the end of introduction.
- The sections 2 and 3 are very complete and all has been explained with good detail, so in my opinion they are fine and adequate and no changes are needed.
Thank you. Based on other reviewers’ comments, sections 2 and 3 were made more neat and concise but still maintaining the same level of details.
- In relation to sections 4, 5 and 6, the description of results is very detailed, however their discussion should be improved. I suggest to include a new section, for example after section 6, only dedicated to discuss the results. With this, the discussion will be highlighted in the manuscript and easier to understand by the readers.
Detailed discussions were added in each section, and the results are summarized in the conclusion instead of a separate section.
- Regarding the conclusion section, I think that it is adequate. I like the idea of using bullet points for emphasizing the main conclusions of the work. This make clearer the relevant findings of the research included in the manuscript.
The bullet points remain following your comments, but the contents are rewritten based on the reviewers’ comments and requests.
Round 2
Reviewer 2 Report
Dear authors,
The way in which the changes made to the manuscript are presented makes it difficult to revise it (it presents both the removed text and the new text, at the same time, in a base version different from the one I reviewed in November 2020).
After a very careful analysis of the submitted revised document (materials-1028937-peer-review-v2), I found that when removing the text marked as deleted, the manuscript is exactly the same as the previous version revised by me in November 2020. Thus, it is impossible to check in the submitted revised manuscript, because they do not exist, the changes that the authors refer in the responses to my previous review comments, since the correct revised version was not submitted.
In summary, I received your responses to my comments, but it was not possible to check the changes made to the manuscript to incorporate these improvements into the article.
For these reasons I maintain the comments made in the previous review.
Author Response
Dear authors,
The way in which the changes made to the manuscript are presented makes it difficult to revise it (it presents both the removed text and the new text, at the same time, in a base version different from the one I reviewed in November 2020).
After a very careful analysis of the submitted revised document (materials-1028937-peer-review-v2), I found that when removing the text marked as deleted, the manuscript is exactly the same as the previous version revised by me in November 2020. Thus, it is impossible to check in the submitted revised manuscript, because they do not exist, the changes that the authors refer in the responses to my previous review comments, since the correct revised version was not submitted.
In summary, I received your responses to my comments, but it was not possible to check the changes made to the manuscript to incorporate these improvements into the article.
For these reasons I maintain the comments made in the previous review.
-->
We submit again a carefully reviewed manuscript and answers to reviewers’ comments. Authors appreciate reviewers’ understand and detailed reviews. The following is the answers to reviewers’ comments for previous round of review process.
---------------------------------------------------------------------------------------
- General comment:
- Most of the information presented (test procedure, results obtained in PLT and LWD and interpretation of these results) has already been published by the authors in Sustainability - MDPI (Assessment of Field Compaction of Aggregate Base Materials for Permeable Pavements Based on Plate Load Tests - 2018, Compaction Quality Monitoring of Open-Graded Aggregates by Light Weight Deflectometer and Soil Stiffness Gauge - 2020). Thus, the information presented in sections 2, 3 and 4 has already been published. In relation to section 5, which presents the use of linear regression and ANN to predict the stiffness of the tested materials, it lacks a more detailed explanation and justification, statistically supported, to effectively prove the validity of Ev1, Ev2 and Elwd predictions
Authors tried and modified to make sections 2, 3, and 4 more concise. In addition, in section 5, detailed discussions were made.
- Specific comments:
Abstract and Keywords
- Abstract needs revision. The connection between the introductory part on permeable pavements in urban areas and what is actually presented in the study is not clear. It is also not completely clear that the study focuses on OGAs materials applied in road pavement base layers and the gap or novelty addressed.
Abstract has been rewritten to make the motivation and novelty clear.
- Keywords – suggestion: include "modulus" and remove "low impact development".
The authors have removed “low impact development” and added “modulus” as suggested.
- Introduction
- This section should include a broader bibliographic review on permeable layers of road pavements.
Literature reviews have been expanded and included in the manuscript.
- Figure 1 must be presented / explained in the body of the text (all figures must be presented in the text – review for Figures 1, 9, 12 and 13).
The authors have included explanations or citations of Figures 1, 9, 12, and 13.
- Field modulus evaluating devices
- Page 2 – lines 72-74: include bibliographic references of the referred standards.
The References for those standards have been added.
- Caption and graphs in Figure 3 are only understood after reading section 3 (field experiment) – improve/review.
The figures are simplified to one figure to help readers understand the concept of the test better and to get rid of complexity, and a corresponding additional explanation is added in those sections.
- Page 3 - line 80: why start with “Here, Elwd is the …"?
Some corrections and improvements in the text have been included to explain LWD and ELWD better.
- Information presented in page 4 – lines 94-107 is not clear – review.
We have improved the sentences.
- Page 5 – line 110 – “…few studies on OGAs to date…” include citations of the few studies performed.
There were minor errors in those sentences. As far as the author’s knowledge, there is no research about the modulus-based compaction quality control of OGAs except for the previous research conducted by the authors, and most studies regarding it have been focused on conventional road base materials. The sentence has been changed to express this aspect in the manuscript.
- Table 1 – units are missing – review.
The units of all the values in Table 1 are checked and presented properly.
- Figure 6 and 7 - Include the letters (a), (b), (c) and meaning in caption. Figure 8 – correct citation [29].
Descriptions for (a), (b), and (c) has been added in Figure 7, and citation has been corrected in Figure 8.
- Comment on the materials studied in view of the standards limits presented. Clarify why these 5 materials (and not others) were studied.
The particle sizes of those 5 materials approximately lie in a commonly used particle size range of the base materials of permeable pavements. This reason has been included in the manuscript.
- Table 2 can be removed, since it repeats the information presented in the text.
The authors tried to make the manuscript concise and neat overall. In the case of Table 2, there was a request of another reviewer and the authors keep this table in the manuscript.
- An area of 4.0m x 2.1m for each type of material is representative and is the spacing between the 24 tests carried out in this area sufficient to guarantee reliable results - not influenced by the tests carried out in the vicinity? Include a comment on these aspects.
LWD is non-destructive test and therefore this can be applied near the test of its own kind or other type. PLT, however, involved high deformation and therefore it is important to make sure they do not interfere each other. The size of plate of PLT test is 30 cm, and each of PLT is at lease 30 cm, apart from another. Considering, the shape of pressure bulb, and the dimension, and also highly localized shape of deformation, each PLT would not affect another.
- Modulus of open-graded aggregate
- Page 7 – line 161: “80 types of materials” or “80 cases”? (16 cases per type of material)
Thank you for pointing that out. The sentence has been improved.
- Include in the text a brief explanation of the following aspects: what Ev1, Ev2 and Elwd represent? Clarify what the first lift, second lift, area A and area B columns represent.
Explanations for Ev1, Ev2, Elwd have been improved or added in section 2. In addition, descriptions for area A and B have been improved and incorporated in the manuscript.
- Discussion on Ev1, Ev2 and Eldw increase rates should be presented and supported by values clearly visible in graphs. The 3D graphs in Figure 11 are not perceptible. Please consider for better interpretation the use of colours or the preparation of line graphs (already presented in a previous publication of these results)
The picture has been modified.
- Variation of Elwd results with the number of roller passes should be improved.
Following the comments, the moduli versus the material type and number of roller passes are newly drawn with rewritten texts in section 4.2.
- Also consider to include a comment on the probable causes related to results trend deviations (proximity of tests, mixture of aggregates?)
The comments on field compaction and differences of measuring devices in the field have been elaborated in the manuscript. “In addition, the experimental study conducted in lab environment does not necessary reflect what would happen in the field. The magnitude of compaction energy and the way that the energy is delivered cannot be the same in the field and lab. In this study, the moduli of open-graded aggregates of various sizes under various compaction energies are investigated based on PLT (Plate Load Test) LWD (Lightweight Defectometer). There has been very few experimental study on the stiffness of open-grade aggregates in field condition.” “PLT costs more much more time and efforts to conduct that LWD, and replacing PLT by LWD would make quality assessment procedure simpler. The intention of models ANN2 and LR2 are, to predict the results of PLT based on those of LWD.” “During the first loading, the specimens are compressed and densified (or compacted), and therefore the second compression curve (or Ev2) is less sensitive to the number of roller passes. The first loading of plate load test also involves significant shear displacement, which does not happen during the second static loading or dynamic loading; Ev1 is more sensitive to the number of roller passes than Ev2 and ELWD overall.”
- Linear regression and ANN using matlab
- Support the information presented on ANN and linear regression with citations and references.
Reference and citations have been added [41-44, 47].
- Why not consider an ANN A3 model with M1, N2, Ev1 and Ev2 inputs and Elwd output? The same for linear regression, ie, why was not considered an equation for Elwd as a function of M1, N2, Ev1 and Ev2?
PLT costs more much more time and efforts to conduct that LWD, and replacing PLT by LWD would make quality assessment procedure simpler. The intention of models ANN2 and LR2 are, to predict the results of PLT based on those of LWD. While the models ANN1 and LR1 employs only the material type and number of roller passes as inputs, the models ANN2 and LR2 has additionally ELWD as an input. It is elaborated and added in the manuscript.
- What specific input information about the tested materials (M1) is used in the AAN models and linear regressions? This aspect must be clearly presented in the text.
Information about the values of M1 has been added in the manuscript. Note that the term M1 has been changed to Nmat.
- The fit of the obtained models to the real data must be analysed based on the interpretation of statistical tests that allow to evaluate this aspect (for example, in the linear regressions through the R2-value and the standard error of the estimates).
The authors have included RMSE analysis in Tables 4 and 5 and elaborated them in the manuscript to reflect the reviewer’s comment.
- Page 12 - line 226-227: taking into account that 5 materials were tested and 4 cases of compaction were considered, how were the percentages defined and cases assigned for training, testing and validation? Do the authors mean 22% instead of 12%?
The explanation about the reviewer’s comment has benn included in section 5.2 “The experimental data had total 80 sets as presented in Table 3, and 70%, 15%, and 15% of data were randomly selected and used for training, validation, and testing (prediction), respectively, with an option implemented in MatLab.”
- Page 13 – line 234: Author’s state that the results obtained in ANN and linear regression predictions are not significantly different - What supports this statement? Clarify.
Authors elaborated it and presented in the paper. “Table 4 presents the RMSE values for training and testing of ANN1 and LR1 (see Figure 13). In case of the model LR1, for all three moduli ELWD, Ev1 and Ev2, the RMSE values for training (fitting) and testing (prediction) are quite close; during fitting and evaluation the LR1 model produces similar level of errors. The model ANN1, on the other hand, for ELWD and Ev2, the RMSE for testing is larger than that for testing. When the testing RMSEs of two different models ANN1 and LR1 are compared, RMSE of ANN1 is higher for ELWD, but RMSE of LR1 is higher for Ev1. One model is not necessarily superior to the other for prediction. The results of the model evaluations of ANN2 and LR2 (see Figure 14) are presented in Table 5. The ANN2 model works better for training (or fitting), but the LR2 model is slightly better at testing (prediction).”
- Figure 15 - why are only 12 cases presented? These are the verification cases?
From 80 sets of data, 56 (70%) of them were used as model training, 12 (15%) of them were used as model validation which is a step for improving the model, and the other 12 (15%) were used as a prediction, whose results are presented in Figure 15 and 16. The authors have made some changes in the text for improving clarity.
- Are RMSE values of 17 or 23 acceptable? Support / Justify.
RMSEs of Ev2 are high because the values of Ev2 are larger than the values of Ev1 and ELWD. There is no criterion or threshold, but it seems the presented models reasonable well follow the baselines as presented the figures where the comparions of the predictions and the baselines are made.
- Clarify whether the models, equations and RMSE values were obtained with all cases (5 types of materials and 4 types of compaction), training representative data or other set of data. This aspect is not clearly presented in the manuscript.
Section 5 is rearranged and rewritten to present them better following reviewers’ comments.
- Figures
- Correct numbering of Figures (there are two Figures 12).
Numberings of Figures have been corrected.
- Conclusions
- Conclusions should be reviewed taking into account the aspects to be improved, especially the ones pointed to section 5.
The conclusion has been completely rewritten and presented in the manuscript.
- Clearly state the scientific gap or novelty addressed with the results obtained.
The behavior or mechanically properties of dense granular materials have been widely studied worldwird. However, open-graded aggregates has not been studied much; there are very few. This study is one of few studies (if not the only) on OGA especially in field scale. Authors pointed it out in the paper. “In addition, the experimental study conducted in lab environment does not necessary reflect what would happen in the field. The magnitude of compaction energy and the way that the energy is delivered cannot be the same in the field and lab. In this study, the moduli of open-graded aggregates of various sizes under various compaction energies are investigated based on PLT (Plate Load Test) LWD (Lightweight Defectometer). There has been very few experimental study on the stiffness of open-grade aggregates in field condition.”
- Include future work to be developed - for example: in addition to the mechanical characteristics of the OGAs, the characterization of their permeability can support the decision on the best material to use in permeable pavements base layers.
OGA materials are highly porous (40%) and the size of pore is also very large. As such, when the water is applied on compacted OGA, it just immediately drains and the concept or definition of permeability can not apply. It is not laminar flow (basis for Darcy’s law) at all.

Reviewer 3 Report
Thanks to the authors' hard work. Almost all my comments are revised. Only one comment:
1. In Nederland, almost all the pavement are porous pavement, include high traffic load highway and low traffic load local connections.
Author Response
Thanks to the authors' hard work. Almost all my comments are revised. Only one comment:
- In Nederland, almost all the pavement are porous pavement, include high traffic load highway and low traffic load local connections.
-->
It is mentioned in the introduction. We appreciate reviewers’ valuable comments.
